

# Effects of inconsistent reporting, regulation changes and market demand on abundance indices of sharks caught by pelagic longliners off southern Africa

Gareth L. Jordaan[1], Jorge Santos[2] and Johan C. Groeneveld[1,3]

[1] Oceanographic Research Institute, Durban, South Africa
[2] Norwegian College of Fishery Science, University of Tromsø, Tromsø, Norway
[3] School of Life Sciences, University of KwaZulu-Natal, Pietermaritzburg, KwaZulu-Natal, South Africa

## ABSTRACT

The assumption of a proportional relationship between catch-per-unit-effort (CPUE) and the abundance of sharks caught by pelagic longliners is tenuous when based on fisher logbooks that report only retained specimens. Nevertheless, commercial logbooks and landings statistics are often the only data available for stock status assessments. Logbook data collected from local and foreign pelagic longline vessels operating in four areas off southern Africa between 2000 and 2015 were used to construct standardized CPUE indices for blue sharks *Prionace glauca* and shortfin makos *Isurus oxyrinchus*. Generalized linear mixed models were used to explore the effects of year, month, vessel, fleet and presence of an observer on blue shark and shortfin mako variability. Landing statistics and auxiliary information on the history of the fishery, regulation changes, and market factors were superimposed on the CPUE indices, to test hypotheses that they would influence CPUE trends. Indices in the West and Southwest (Atlantic) areas were elevated for both species, compared to the South and East (Indian Ocean). The scale of year-on-year CPUE increments, up to an order of magnitude for blue sharks, reflected occasional targeting and retention, interspersed with periods where blue sharks were not caught, or discarded and not reported. Increments were smaller for higher value shortfin makos, suggesting that indices were less affected by unreported discarding. CPUE indices and landings of both shark species have increased in recent years, suggesting increased importance as target species. Analysis of logbook data resulted in unreliable indicators of shark abundance, but when trends were interpreted in conjunction with landings data, disaggregated by area and month, and with hindsight of market demand and regulation changes, anomalies could be explained.

# INTRODUCTION

Catch-per-unit effort (CPUE) trends based on fisher logbooks and landings data are widely used in fisheries research to estimate the relative abundance of fished stocks. The key assumption that abundance is proportional to CPUE is rarely met, however,

Corresponding author
Gareth L. Jordaan,
gjordaan@ori.org.za,
jordaan.gareth@gmail.com

because the relationship is affected by intrinsic catchability fluctuations of fished stocks, and also by variability in fishing methods used and gear characteristics. *Maunder & Punt (2004)* reviewed methods to standardize catch and effort data to account for the added variability in inter-annual CPUE trends. Apart from variability that can be addressed within a standardization framework, underreporting of catches will affect the data from which abundance indices are derived (*Groeneveld, 2003*; *Rudd & Branch, 2017*; *Van Beveren et al., 2017*). For example, unwanted sharks that are discarded overboard by pelagic longliners targeting mainly tunas and swordfish are infrequently reported in logbooks, and landing statistics therefore underrepresent the actual numbers of sharks caught during a fishing trip (*Campana, 2016*).

Standardized CPUE trends nevertheless remain an important relative measure of the abundance of sharks caught by pelagic longline gear in oceanic habitats. They are used by tuna Regional Fisheries Management Organizations (tRFMOs), such as the Indian Ocean Tuna Commission (IOTC) and the International Commission for the Conservation of Atlantic Tunas (ICCAT) to assess the status of sharks caught as a bycatch of tunas and swordfish (*Francis, Griggs & Baird, 2001*; *Su et al., 2008*; *Petersen et al., 2009*; *Cortés, 2013*; *ICCAT, 2016*; *IOTC, 2016*). Some studies have used auxiliary information collected by fisheries observers at sea to assess discards (*Cortés, 2013*) but despite these efforts, most shark abundance indices retain high uncertainty.

Pelagic sharks are vulnerable to fishing pressure, because of their life history traits and behaviour patterns (*Dulvy et al., 2008*). As a group, they have lower productivity than teleosts, are long-lived and slow-growing, and produce few offspring which mature late (*Musick, 1999*). Pelagic sharks are predatory animals, and some species are found in association with other target species of longline gear, on which they prey or compete with for food (*Mejuto, García-Cortés & Ramos-Cartelle, 2008*). Inevitably, these sharks are caught in large numbers in areas of intensive longline fishing (*Campana, 2016*). Pelagic sharks migrate freely and widely over their range, often across international boundaries (*Kohler et al., 2002*; *Block et al., 2011*; *Campana, 2016*), thus making them vulnerable to high seas fishing fleets.

Whether a captured shark will be discarded or retained and processed depends primarily on species, as a proxy for economic value, and the regulatory environment within which a fishery operates (*James et al., 2016*). Commonly retained sharks with high quality meat and marketable fins include shortfin makos *Isurus oxyrinchus*, whereas blue sharks *Prionace glauca* are less valuable, and are more often discarded. Blue sharks dominate the bycatch of pelagic longline fisheries in subtropical and temperate waters worldwide (*Oliver et al., 2015*), and exceed the catches of tuna and swordfish target species in some areas (*Campana, Joyce & Manning, 2009*). They are faster-growing and relatively more productive than most other shark species, and considered less vulnerable to fishing pressure (*Aires-da Silva & Gallucci, 2007*). Shortfin makos make up a large proportion of retained bycatch of longline fisheries, and sometimes form the target of shark-directed fisheries (*Francis, Griggs & Baird, 2001*; *Campana, Marks & Joyce, 2005*; *Petersen et al., 2009*; *Bustamante & Bennett, 2013*). Makos are characterized by low population growth rates, and are more vulnerable to overfishing than blue sharks (*Dulvy et al., 2008*).

Neither species is considered to be in imminent danger of collapse, although populations of one or both might be overexploited (*Campana, Marks & Joyce, 2005*; *Campana, 2016*). Reported catches of blue sharks in the Indian Ocean have continued to increase since the early 1990s (*IOTC, 2016*), although catch rate estimates are highly uncertain, and probably represent only the sharks that were retained onboard. North Atlantic stocks of shortfin makos appear to be overfished and undergoing overfishing, whereas the status of South Atlantic stocks are highly uncertain (*ICCAT, 2017a*; *ICCAT, 2017b*). The abundance of shortfin makos in the Indian ocean appear to have increased in recent years (*Kimoti et al., 2011*; *Coelho, Infante & Santos, 2013*), but data may have been affected by changes in reporting practices, rendering estimates unsure (*Hoyle et al., 2017*). Overall, shark landings reported to the FAO peaked in 2003, and in the decade since then have declined by almost 20%—a reflection of overfishing, rather than good management (*Davidson, Krawchuk & Dulvy, 2016*).

Both distant-water pelagic longline fleets and local vessels fish for tunas, swordfish and sometimes sharks around southern Africa, where the Southeast (SE) Atlantic and the Southwest (SW) Indian Oceans meet (*Petersen et al., 2009*; *Da Silva et al., 2015*). Early local longline fisheries for tunas (1960s) and sharks (since 1992) in South African waters met with varied success (*BCLME, 2005*; *Petersen & Goren, 2007*). Japanese and Korean flagged vessels were licensed to fish in South African waters from the early 1990s, where they set deep longline gear to target tunas (*Petersen & Goren, 2007*). Thirty experimental tuna permits were issued to South African-flagged vessels in 1997, and the local fishery was formalized in 2005, when long-term fishing rights were issued respectively for swordfish- and tuna-directed fisheries, and the shark-directed permits were abolished. Sharks continued to be targeted by some local vessels under a permit exemption, but their licenses were fully amalgamated into the tuna and swordfish fishery in 2011 (*Da Silva et al., 2015*), whereafter sharks were considered as a bycatch.

*Petersen et al. (2009)* reported that blue sharks and shortfin makos were the most common sharks caught in longline gear set around southern Africa, and that local swordfish-directed vessels caught more sharks than foreign vessels. Based on data collected by independent observers onboard vessels, they observed declining abundance of both shark species between 2002 and 2007, accompanied by a decline in average shark size. Data collected by fisheries observers at sea are rarely available for long uninterrupted periods, and therefore long-term indices of abundance must often rely on logbook and landings data—even though these data may be affected by targeting practices and under-reporting.

*Campana (2016)* highlighted the risks associated with ineffective fisheries management structures, unmonitored fishing mortality (particularly of discarded sharks), and the scarcity of basic information on the status of shark populations. Importantly, species-specific patterns and information gaps, by ocean region and individual fishery, have been identified as critical weaknesses in terms of global conservation and improved fisheries management efforts (*Oliver et al., 2015*). We used logbook data collected from local and foreign pelagic longline fleets between 2000 and 2015 to construct standardized CPUE trends of blue sharks and shortfin makos reported in four fishing areas straddling the ICCAT and IOTC reporting regions. Official landing statistics and auxiliary information

 

on reporting practices, changes in the regulatory environment and market demand were superimposed on the CPUE trends, to test hypotheses that they would coincide with abrupt changes in the CPUE indices. Our study provides a reference framework for the analysis and interpretation of commercial logbook data affected by unreported discards—often the only source of information that fisheries researchers have for assessments.

## MATERIALS AND METHODS

### Study area

The study was undertaken in the SE Atlantic and SW Indian Oceans, adjacent to the coast of South Africa. Longlines set within the Exclusive Economic Zone (EEZ, within 200 nautical miles from the shore) and in the surrounding high seas were included. The study area straddled the ICCAT (west of 20°E) and IOTC (east of 20°E) reporting regions (Fig. 1). The ICCAT region along the west coast of southern Africa is dominated by cool-temperate waters and productive upwelling systems of the northwards flowing Benguela Current (*Hutchings et al., 2009*). The IOTC region along the east coast is influenced by the western boundary Agulhas Current, which brings warmer, less-productive subtropical waters southwestwards (*Lutjeharms, 2006a*; *Lutjeharms, 2006b*; *Beal et al., 2011*). The boundary zone between the two systems off the southern tip of Africa is highly dynamic, influenced by current strength and direction, bottom topography, seasonality and climatic events (*Swart & Largier, 1987*).

The study area was stratified into four areas, to reflect the tRFMO boundary and incorporate known oceanographic features, as follows: West (a cool temperate area influenced by the Benguela Current, extending from the Namibian border to 33°S); Southwest (a dynamic boundary zone, which includes the western Agulhas Bank, between 33 and 20°E); South (the lower Agulhas Current area, where the narrow shelf broadens towards the west to form the eastern Agulhas Bank, between 20 and 26°E); and East (subtropical waters influenced by the upper Agulhas Current, from 26°E to the Mozambique border (Fig. 1)). The four areas were similar to those used in a previous study on pelagic shark bycatches (*Petersen et al., 2009*).

### Data and assumptions

Logbook records completed on a set-by-set (daily) basis were obtained from the Department of Agriculture, Forestry and Fisheries (DAFF) for local South African-flagged vessels and foreign-flagged vessels (mainly Japanese and Korean) licensed to fish for tunas in the South African EEZ. All catches were landed in local ports, and logbook records of foreign vessels were available for the 2000–2015 period, excluding 2006, when they did not fish in the region. Records for local vessels were obtained for fishing sets targeted at tunas, swordfish and pelagic sharks between 2000 and 2015. The data comprised of daily retained catches by species by numbers, weight (after reconciling estimated weight at sea with landed weight in port), numbers of hooks set, set and haul positions and times, target species, depth of sets and bait type, and whether a fisheries observer was present or not.

The logbook data were cleaned by removing anomalous records in which setting positions, date, depth, fishing effort (number of hooks), set durations or catch composition

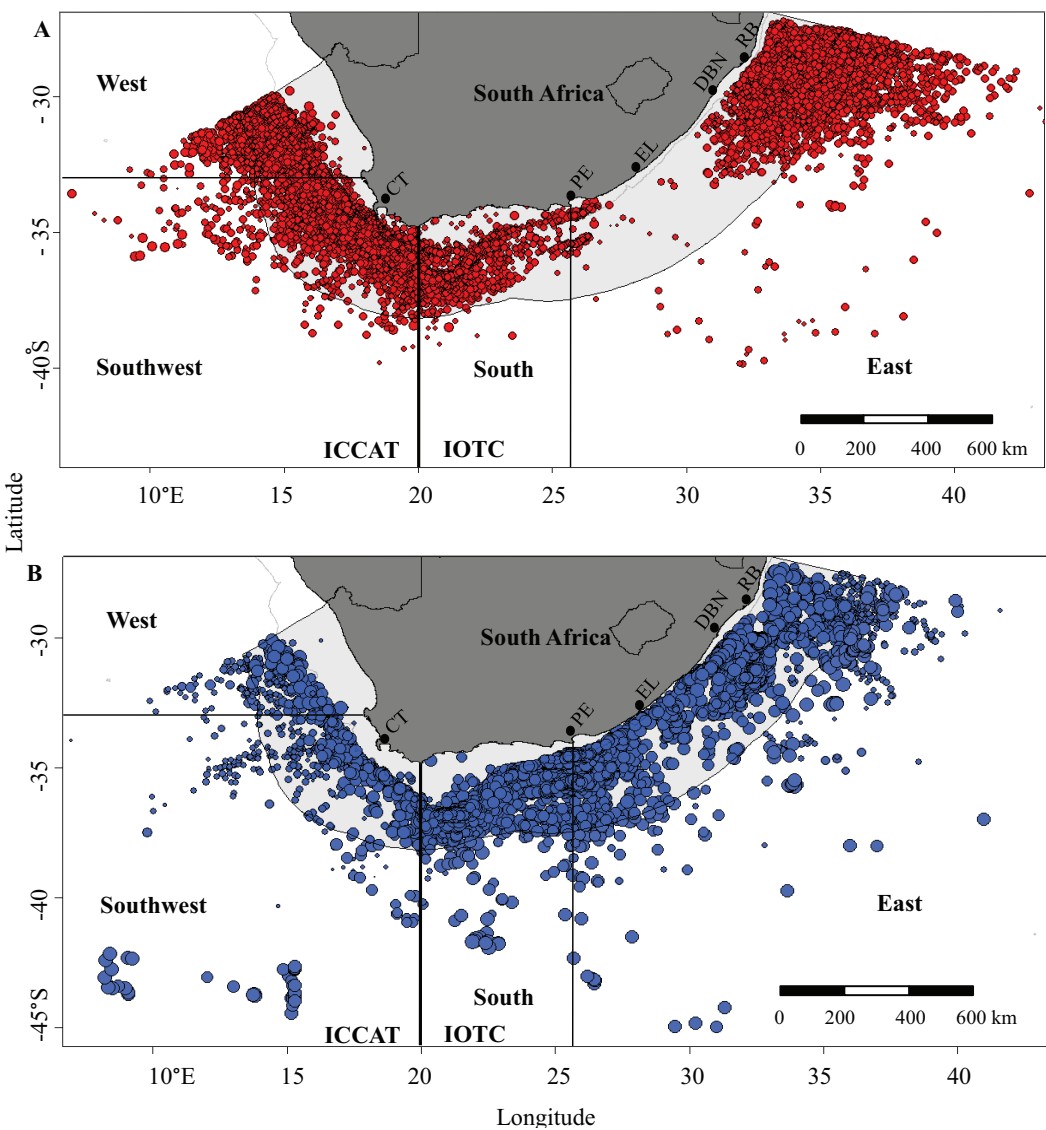

**Figure 1** Geographical distribution of fishing effort by (A) local and (B) foreign pelagic longliners between 2000 and 2015, based on logbook data provided by vessel skippers. The sampling area is subdivided into the SE Atlantic (reporting to ICCAT) and the SW Indian Ocean (reporting to IOTC) along 20°E, and the West, Southwest, South and East sampling areas are shown. Bubble size is proportional to the numbers of hooks set per line. CT, Cape Town; PE, Port Elizabeth; EL, East London; DBN, Durban; RB, Richards Bay.

or quantities were clearly incorrect or mismatched. Records with setting positions outside the SW Indian Ocean and SE Atlantic were removed. Data selected for the study were dates between 2000 and 2015; fishing effort of 310–3,800 hooks per line; a maximum of 801 blue or shortfin makos reported per set; and fewer sharks (both species combined) than hooks per set. Spatio-temporally explicit catch and fishing effort of 29,018 individual sets remained (86% of initial records), and were used for analyses. For the local fleet, 37% of 16,810 sets by 61 vessels recorded no blue sharks and 32% recorded no shortfin makos.

**Table 1 Explanatory variables hypothesised to affect the CPUE of blue sharks and shortfin makos caught by local and foreign pelagic longline fishing vessels from 2000 to 2015.** Models were constructed individually for the West, Southwest, South and East fishing areas.

| Variable | Type and effect | Description |
|---|---|---|
| Year | Categorical, fixed | 2000–2015 (16 levels) |
| Month | Categorical, fixed | January–December (12 levels) |
| Vessel | Categorical, random | 61 individual local-, and 49 foreign vessels |
| Fleet | Categorical, fixed | Local, Foreign (2 levels) |
| Observer | Categorical, fixed | Yes, No, Unknown (3 levels) |

For the foreign fleet of 49 vessels, no blue sharks were reported in 29%, and no makos in 40% of 12,208 sets.

We assumed that blue sharks and shortfin makos were correctly identified in logbooks—these two species are commonly caught and relatively easy to identify. Longfin makos (*Isurus paucus*) have rarely been reported from the sampling region (*Reardon, Gerber & Cavanagh, 2006*), and hence all makos were assumed to be *I. oxyrinchus*. Other shark species made up only a small part of the retained catch, and were mostly grouped in logbook records, as requiem sharks (mostly *Carcharhinus* spp.), threshers (*Alopias* spp.), hammerheads (*Sphyrna* spp.) or as unidentified sharks.

## Data analysis

Variability in blue shark and shortfin mako CPUE (numbers/1,000 hooks) by year, month, vessel, fleet, and observer presence (Table 1) was explored using Generalized Linear Mixed Models in the statistical software package R, version 3.3.2 (*R Development Core Team, 2016*). The R-libraries "lme4" (*Bates et al., 2016*) and "lmtest" (*Hothorn et al., 2015*) were used to run the GLMM procedure. Models were constructed individually for blue sharks and shortfin makos caught in each of the West, Southwest, South and East areas. The distribution of data among treatment cells are shown in Table S1.

The large proportion of zero catches justified the use of the delta method for analysis (*Pennington, 1983*; *Maunder & Punt, 2004*; *Lauretta, Walter & Christman, 2016*). In the first submodel, the probability of a non-zero catch was modelled, based on presence/absence information, and assuming a binomial error distribution. In the second submodel, the positive catch numbers were modelled using a gamma continuous probability distribution. The gamma distribution was chosen because the relationship between the logarithms of the mean and variance of non-zero CPUE records was close to two (data highly dispersed) (*McCullagh & Nelder, 1989*; *Stefánsson, 1996*), and the gamma provided better fits than inflated discrete distributions in preliminary tests.

Final models were selected based on a stepwise approach, involving modelling combinations of error structures, link functions and explanatory variables. The most parsimonious models were selected based on the lowest value of the Bayesian Information Criterion (BIC), and visual assessment of residual plots. Fixed and random effects models provided similar fits, and we selected fixed effects for all variables, except for a random effect for vessel.

Standardized CPUE trends by species and year for each area were computed as the product of the probability of catch (binomial model) and positive catch (gamma model) obtained from the model coefficients. The expected values in the final models were the specific CPUE using year as the fixed (main) effect, as this is the factor of interest with regard to abundance trends (*Maunder & Punt, 2004*; *Venables & Dichmont, 2004*).

Error back-calculation and error propagation were calculated using the conservative procedures recommended by *Jørgensen & Pedersen (1998)*, *Lindberg (2000)* and *Tellinghuisen (2001)*. This included the back-transformation of coefficients and their errors from logarithmic and logit scales to the observation scales, as well as the propagation of error in models where back-transformed estimates from gamma and binomial models had to be multiplied.

Anomalies in the inter-annual CPUE indices were identified *post hoc*, as sudden increments in their scale that could not plausibly be explained on biological or population dynamics grounds alone. Spatio-temporal trends in shark landings, as well as published and anecdotal auxiliary information on the history of the fishery, regulatory framework, and market demand were superimposed over the CPUE indices to evaluate the effects of changes in fishing practices on the observed trends.

## RESULTS

### Nominal trends in fishing effort, landings and shark species composition

Some 52 million hooks were deployed by 110 vessels between 2000 and 2015, 31 million by foreign vessels targeting tuna and 21 million by local vessels targeting mostly swordfish and tuna. The combined fishing effort was greatest in 2011, with a total of nearly 6 million hooks set in that year (Fig. 2A). Foreign vessels deployed an average of $2.0 \pm 1.1$ million hooks per year, compared to $1.3 \pm 0.4$ million by local vessels over the same period. The numbers of hooks set by foreign vessels peaked between May and October each year, whereas local vessels fished throughout the year, with marginally fewer hooks set in January and February than other months (Fig. 2B). Foreign vessels ventured further southwards than local vessels, which tended to remain within the EEZ (Fig. 1).

Foreign fishing effort increased from 0.5 million hooks in 2001 to a maximum of 3.9 million in 2011, when 15 vessels were active; thereafter it declined to 0.8 million hooks set by four vessels in 2015 (Fig. 2). Zero effort was recorded in 2006, when no foreign vessels were licensed to fish for tunas in the South African EEZ. Local fishing effort increased gradually between 2004 and 2011, thereafter remaining stable at 1.5–1.9 million hooks set per year by 15–17 vessels up to 2015.

Local vessels fished in all four areas, but in the East their range was limited to the northern half of the area, near a landing site at Richards Bay (Fig. 1). Foreign vessels fished mainly in the SW Indian Ocean, with the bulk of all hooks set in the South (58%) and East (33%) areas, and the remaining 9% in the SE Atlantic. Foreign vessels set an average of $2,493 \pm 597$ (SD) hooks per line, compared to only $1,282 \pm 250$ hooks per line used by local vessels.

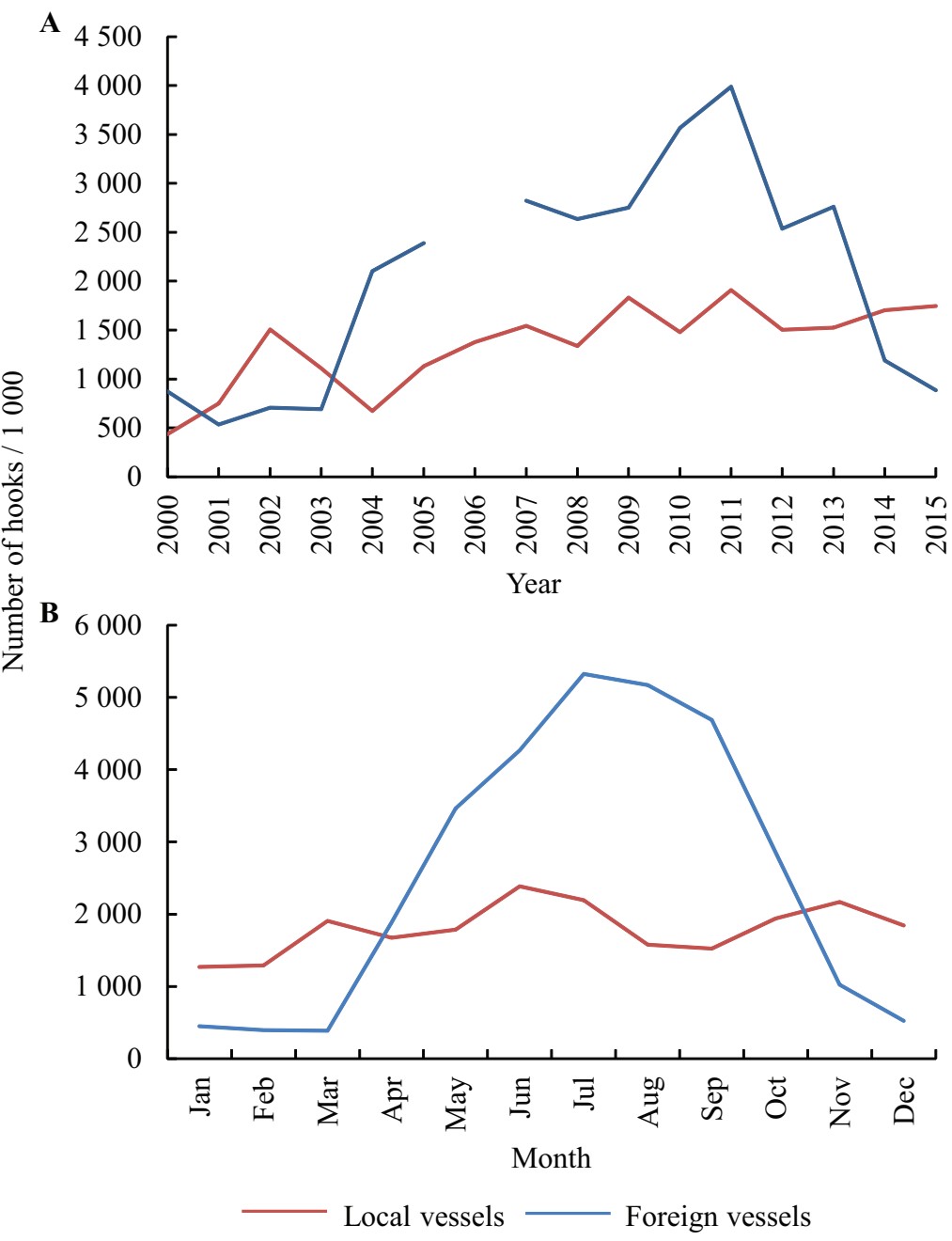

**Figure 2 Numbers of hooks set per (A) year (2000–2015) and (B) per calendar month, as reported by local and foreign pelagic longliners fishing in the study area.** The distribution of the data by area and season is shown in Table S1.

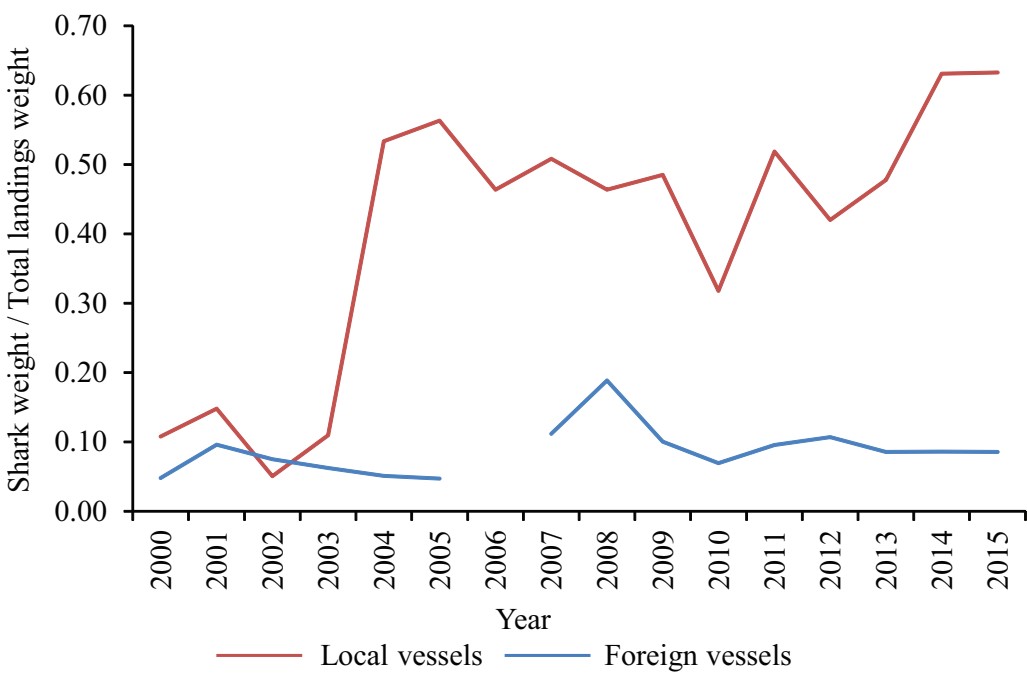

**Figure 3** Ratio of shark landings (weight of blue and shortfin mako sharks combined) to total landings (all species, including tunas, swordfish, blue sharks and shortfin makos) between 2000 and 2015, for local and foreign pelagic longliners, respectively.

The ratio of shark landings to total landings by weight (all species, including tunas, swordfish, blue sharks and shortfin makos) remained <0.2 for both fleets between 2000 and 2003, whereafter it increased sharply for the local fleet, to >0.5 in 2004 (Fig. 3). The local fleet ratio remained relatively constant at this higher level up to 2013, and then increased to the highest level on record (>0.6) in 2014 and 2015. With few exceptions, the ratio of sharks caught by the foreign fleet remained at or below 0.1 in all years.

A total of 681,456 sharks (10,070 t) were reported by local and foreign fleets combined between 2000 and 2015. Blue sharks dominated shark landings by numbers (%N = 58%) whereas shortfin makos contributed most to landed weight (%W = 59%). 'Other sharks' could not confidently be resolved to species level in the landings, but they made up only 1% N and 3% W. Nearly 4,000 sharks were not identified beyond 'Sharks nei' (not elsewhere included), 3,500 were grouped as requiem sharks (including several similar-looking Carcharhinid species, but excluding blue sharks), and neither thresher sharks (*Alopias* spp.) nor hammerheads (*Sphyrna* spp.) were identified beyond genus level. Tope sharks (*Galeorhinus galeus*) (included with the requiem shark group) were first identified in landings in 2011.

Local vessels contributed the bulk of all sharks landed, comprising 91% N and 88% W (Fig. 4). By area, sharks were the most common group in landings by local vessels fishing in the Southwest (66% W) and South (79% W) (Fig. 5). Only 32% W of landings from the West were sharks, but they were mostly small sharks, contributing 70% N of landings from that area. Local vessels fishing in the East landed mainly tunas and swordfish, and fewer

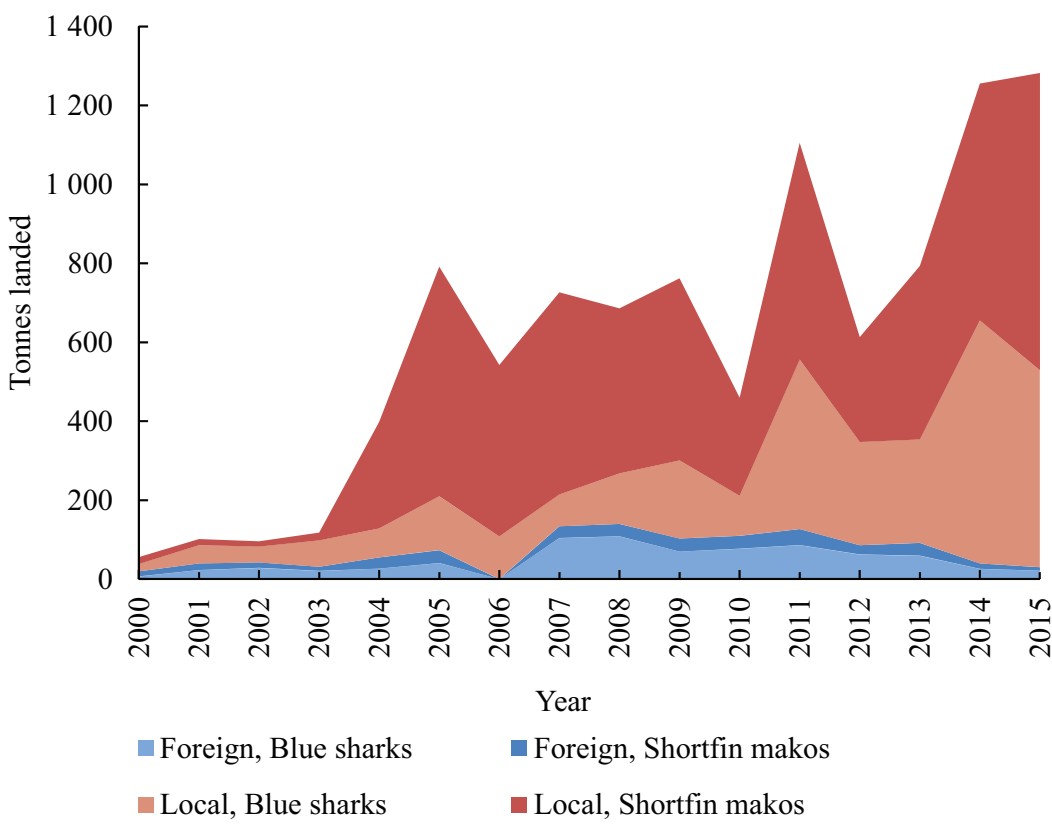

**Figure 4** Cumulative weight of blue sharks and shortfin makos landed per year (2000–2015) by local and foreign pelagic longliners, respectively.

sharks (21% N; 14% W). Sharks contributed <10%W of foreign vessel landings from the South and East (SW Indian Ocean) and around 15% W in the West and Southwest (SE Atlantic) (Fig. 5). Overall, sharks contributed 52%N and 31% W of the combined landings reported by both fleets.

Blue sharks and shortfin makos dominated shark landings (Fig. 5). Prior to 2004, nearly all landed blue sharks were reportedly caught in the West, but thereafter landings from the Southwest and South became progressively more important, especially after 2010, when landings increased substantially (Fig. 6). Blue shark landings peaked at nearly 80,000 individuals in 2014, when most sharks originated from the Southwest. A similar trend occurred in 2011 and 2015, when blue shark landings were also elevated (>50,000 sharks/y).

Shortfin mako landings increased sharply in 2004 and 2005, with most sharks originating from the South during that period (Fig. 6). Mako landings peaked in 2011 (nearly 30,000 sharks), and after a decline, peaked again in 2014 (27,000) and 2015 (38,000). The elevated landings in all three years originated mainly from the Southwest and South areas, with much smaller landings from the West and East.

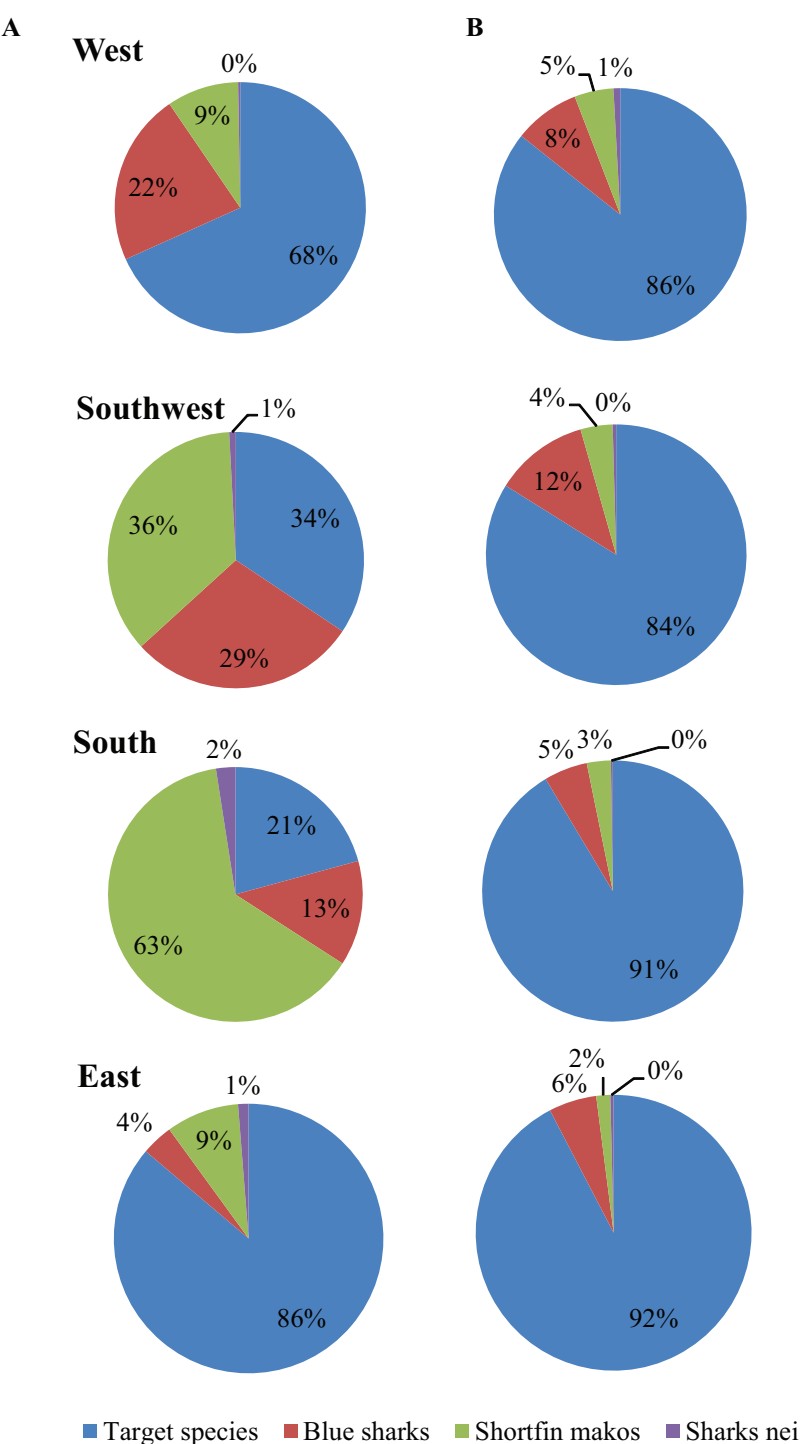

**Figure 5** Trends in the numbers of blue and shortfin mako sharks reported by (A) local and (B) foreign pelagic longline vessels between 2000 and 2015.

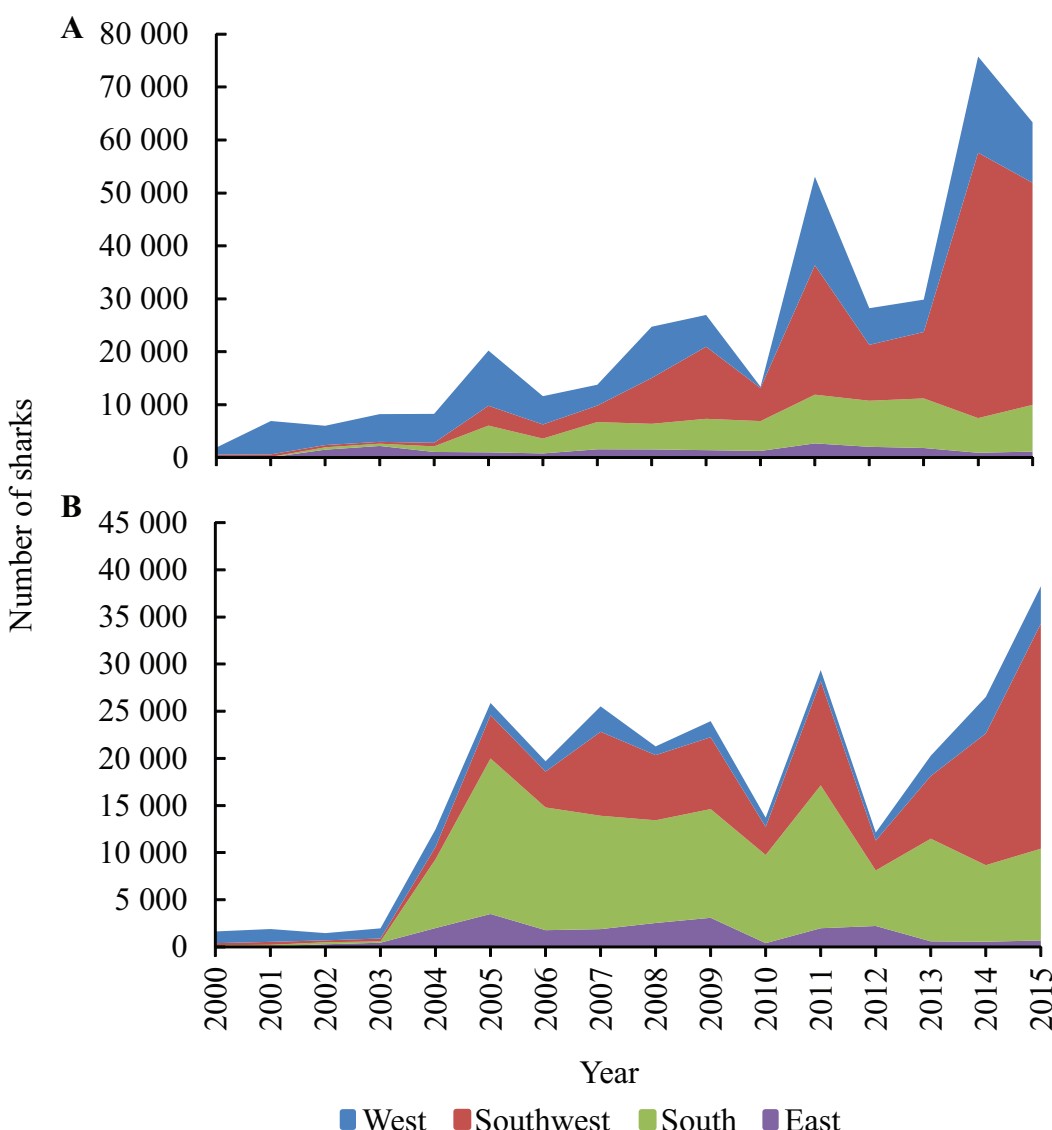

**Figure 6** Cumulative numbers of (A) blue sharks and (B) shortfin makos reported in logbooks per area, between 2000 and 2015.

## CPUE standardization

The final GLMM's selected to model CPUE trends for blue sharks and shortfin makos for each of the four areas included vessel as a random effect and year, month, fleet and the presence or absence of an observer as fixed effects. These combinations consistently provided the best fits. The estimates, standard errors and *p*-values of the binomial and gamma submodels for each area are provided in Tables S2–S5.

For blue sharks, the fleet variable was significant in three of four binomial models and in three gamma models. Based on the coefficients, blue sharks were present more frequently in sets made by the foreign than the local fleet, but when present, far greater numbers of blue sharks were retained by local vessels. Observer was a significant factor in three of
four binomial- and two gamma models. Blue shark presence and numbers reported were consistently higher in the absence of an observer, suggesting increased targeting when not observed. Month was a significant factor in all binomial and gamma models, thus verifying that season affects the availability of blue and shortfin makos.

For shortfin makos, fleet was a significant factor in one of four binomial models and in two gamma models. Shortfin makos were encountered more often, and in higher numbers in sets made by the local- than foreign fleet. The presence of an observer was significant in all four binomial- and three of four gamma models for shortfin makos. As for blue sharks, shortfin mako presence and numbers reported increased when an observer was absent.

Although not consistent, month was often significant in the binomial and gamma models for all four areas, and in most cases the presence and numbers of shortfin makos increased over the autumn and winter months (Tables S2–S5). For shortfin makos, the probability of encounter (binomial) was higher during autumn and winter months, and the numbers per set (gamma) were higher in two areas (Southwest and South) in all four seasons. No consistent seasonal trend was apparent for blue sharks.

The standardized CPUE indices for blue sharks over the 16-year period differed substantially between the four areas (Fig. 7). Prior to 2003, blue shark CPUE was low in all areas, but it increased substantially in 2004 and 2005, relative to initial values. Comparatively larger increases occurred in 2011 (West), 2008–2014 (Southwest), 2008 and 2013 (South) and in 2011 and 2014–2015 (East). The CPUE indices were highly variable between years, especially in the West, where the number of blue sharks/1,000 hooks increased from 4 in 2010 to 191 in 2011, and decreased to 57 two years later, in 2013. Notably, the maximum index values of blue sharks in the West and Southwest were an order of magnitude greater than in the South and East, reflecting their numerical preponderance in landings from these areas (Fig. 6). By weight, blue sharks were also relatively more important in landings, (compared to other species) in the West and Southwest (Fig. 5).

In 2011 and 2012, the probability of encounter (binomial) and numbers per set (gamma) of blue sharks in the West were much higher than in any other year, resulting in exceptionally high peaks in the CPUE index. Blue sharks caught in the West and Southwest were mainly small juveniles, whereas those caught in the other two areas were, on average, much larger and heavier, with their size increasing in an eastwards direction (Fig. 8).

The standardized CPUE indices for shortfin makos were less variable than for blue sharks, with maxima of around 9 sharks/1,000 hooks in the West and Southwest, declining to <2 sharks/1,000 hooks in the South and <1 sharks/1,000 hooks in the East (Fig. 7). Initial increases in shortfin mako CPUE occurred in 2004 in the West, and in 2005 in the other three areas. The standardized trend in the West increased continually from 2008 (<1 shark/1,000 hooks) to 2015 (8 sharks/1,000 hooks), and in the Southwest a similar, but more variable, long-term increase occurred between 2006 (2 sharks/1,000 hooks) and 2015 (9 sharks/1,000 hooks). The CPUE indices in the South and East increased steeply in 2005, but then remained stable or even declined between 2005 and 2015.

For both shark species, the indices suggest a greater abundance in the SE Atlantic (West and Southwest areas) than in the SW Indian Ocean (South and East areas) (Fig. 7). The indices further suggest increasing abundance (or alternatively increasing retention rates)

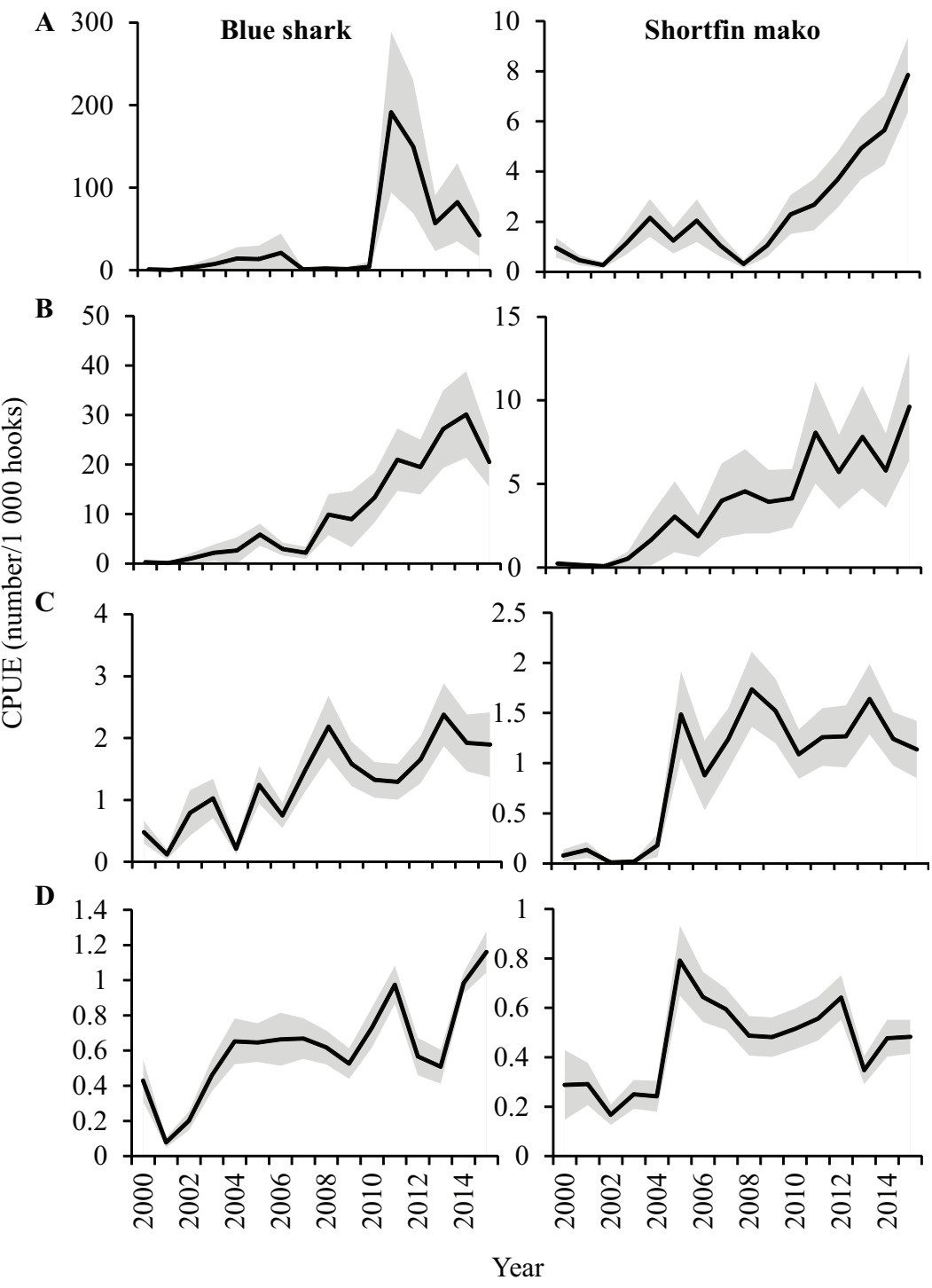

**Figure 7** Standardized CPUE indices (±SE) for blue sharks and shortfin makos in the (A) West, (B) Southwest, (C) South and (D) East areas, based on the binomial and gamma models applied in the study.

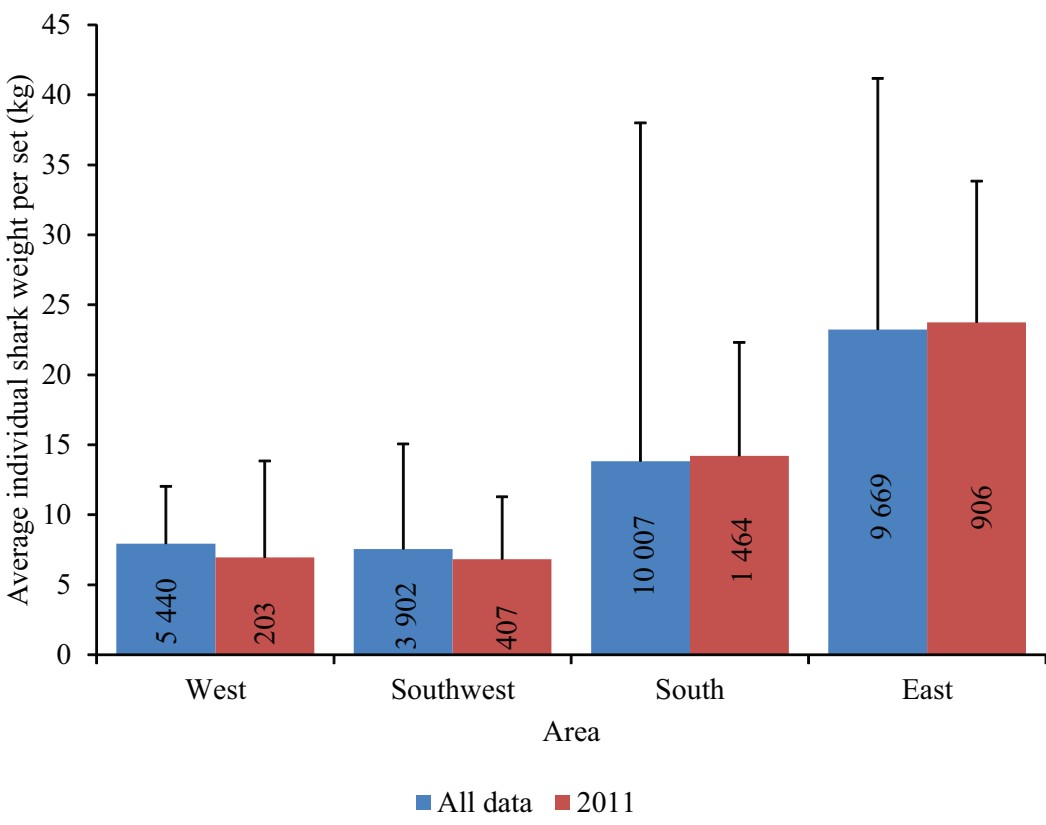

**Figure 8** **The average individual blue shark weight per set (kg) in each area for all years combined, and for 2011 (± SE).** Total number of sets per area (*n*) is included in the bars.

in the SE Atlantic, compared to more stable abundance in the SW Indian Ocean after 2005. The large fluctuations in year-on-year index values in blue sharks in the West suggests that discarding takes place in some years, or that large numbers of small juveniles are sometimes encountered, and retained.

## DISCUSSION

Blue sharks and shortfin makos made up the bulk of sharks landed by the local- and foreign pelagic longline vessels in the present study. Data collected by observers stationed on vessels fishing off southern Africa prior to 2006 also reported blue sharks and shortfin makos as predominant bycatches (*Petersen et al., 2009*). Fleets from Brazil and Uruguay caught mainly blue-, shortfin mako and porbeagle sharks in the South Atlantic (*Hazin et al., 2008*), and Portuguese longliners caught mainly blue sharks as bycatch in the equatorial Atlantic, with shortfin makos and other sharks also present (*Coelho, Santos & Amorim, 2012*). Fleets in the NW Atlantic (*Campana, Marks & Joyce, 2005*; *Campana, Brazner & Marks, 2006*; *Cortés, 2013*), NE Atlantic and Mediterranean (*Megalofonou, 2005*; *Mejuto et al., 2009*), North Pacific (*Walsh & Teo, 2012*) and South Pacific (*Francis, Griggs & Baird, 2001*) also caught mainly blue sharks and varying quantities of shortfin makos. The predominance

of blue sharks and shortfin makos in the present study is therefore consistent with trends from other studies.

Most shark landings made between 2000 and 2015 originated from local vessels (91% $N$; 88% $W$). Proportionally, blue sharks and shortfin makos were more important in local than foreign landings in all four areas, and the combined ratio of shark to total landed weight fluctuated around 0.5 for the local fleet, compared to 0.1 for the foreign fleet. Fishing method, area fished and seasonality of fishing differed between the two fleets, and could partially explain the disproportionate importance of sharks to them. Local vessels set hooks near the surface, overnight, and with gear that included light-sticks and squid for bait. Although directed mainly at swordfish, this configuration also contributes to high shark catch rates (*Stone & Dixon, 2001*; *Dos Santos, Garcia & Pereira, 2002*; *Ward & Myers, 2005*; *Gilman et al., 2008*; *Mejuto, García-Cortés & Ramos-Cartelle, 2008*; *Petersen et al., 2009*). Foreign vessels that targeted tunas used a different gear configuration and set hooks deeper and during daytime, thus avoiding high shark bycatches (*Petersen et al., 2009*). Local vessels fished mostly in the highly productive SE Atlantic and in the transition zone over the Agulhas Bank, where blue sharks and shortfin makos were more numerous (*Petersen et al., 2009*; *Groeneveld et al., 2014*), whereas foreign vessels focussed on the tuna stocks in the SW Indian Ocean.

Blue shark landings originated mostly from the SE Atlantic (West, Southwest) and the South, but the CPUE index was low in the South, compared to the other two areas (see Fig. 7). Similarly, a large proportion of shortfin mako landings originated from the South, but the CPUE index for this area was considerably lower than in the West and Southwest. The anomaly of proportionally high landings but low CPUE in the South was ascribed to an overlap in the fishing areas frequented by local- and foreign vessels in the South. As a consequence, fishing effort in the South was high, but not all of it contributed to shark landings—especially not foreign vessels. The CPUE indices for blue sharks and shortfin makos caught in the South were thus artificially reduced in the model outputs, relative to the Southwest and West areas, where foreign vessels fished less intensively.

The modelled maximum CPUE of 191 blue sharks/1,000 hooks for the West in 2011 was exceptionally high, compared to all other values. A combination of a high probability of encountering blue sharks and large numbers of blue sharks caught per longline set can explain this peak. Given the small size of blue sharks caught in the West (Fig. 8), we suggest that the high index value reflects fishing in a blue shark nursery area, where small juveniles aggregate, and can be caught in large numbers. The retention of large numbers of small juveniles in the West in 2011, potentially as a result of high market demand in that year, can therefore explain the comparatively high numerical index value. In contrast, discarding of blue sharks in other years, without reporting them, would have depressed the index relative to the peak 2011 value, when discarding was presumably reduced.

*Coelho et al. (2018)* suggested that the main nursery grounds for blue sharks in the South Atlantic are in the temperate waters of the SE Atlantic, off western South Africa and Namibia. Furthermore, *Da Silva et al. (2010)* proposed a blue shark parturition and nursery area in the Benguela/Agulhas Current confluence, based on a high frequency of small juveniles in research longline catches. These putative nursery areas overlap the West
and Southwest areas of the present study, and we are therefore confident that the high CPUE index value in 2011 in the West is not unrealistic. Rather, it is indicative of increased retention of small blue sharks caught in nursery areas in some years.

Shortfin makos have a higher economic value than blue sharks (*Dent & Clarke, 2015*), and their CPUE indices are therefore less likely to be affected by discards than in the case of blue sharks. Inter-annual fluctuations are thus smaller, and potentially reflect variations in shortfin mako abundance better. Nevertheless, increased targeting and retention undoubtedly played a major role in 2004 and 2005, when shortfin mako CPUE indices first increased substantially. We suggest that the continued increase in the West (after 2008) and Southwest (after 2006) reflected increased targeting by local vessels, possibly combined with higher abundance. Shortfin mako landings reported to ICCAT by the local longline fleet increased sharply from 250 t in 2013, to 476 t in 2014, and to 613 t in 2015, thus highlighting their growing importance on markets. CPUE indices of other fleets in the SE Atlantic that report to ICCAT also suggested a relative increase in shortfin mako abundance, particularly since 2004, although estimates were highly variable between years and fleets (*ICCAT, 2017b*). Japanese and Portuguese longline fleets in the Indian Ocean have reported increasing shortfin mako abundance in recent years (*Kimoti et al., 2011*; *Coelho, Infante & Santos, 2013*), but trends may have been affected by changes in reporting practices (*Hoyle et al., 2017*).

Most shortfin mako landings originated from the Southwest (SE Atlantic) and South (SW Indian Ocean) areas, at the confluence of the Benguela and Agulhas Current systems, suggesting that they are more abundant there than in the East. *Groeneveld et al. (2014)* found fewer, but larger-sized shortfin makos in the East area, with mean shark size declining from east to west. The smallest individuals, in larger numbers, occurred near the Agulhas Bank edge (South and Southwest areas in the present study) in June to November, suggesting that juveniles congregate there to feed. A juvenile feeding ground at the Agulhas Bank edge could therefore explain why the majority of shortfin mako landings originate from the South and Southwest areas, and also why the CPUE index reached its highest levels in the Southwest.

The market price for shortfin makos increased between 2003 and 2004 (*Dent & Clarke, 2015*; *Autoridad_Portuaria_de_Vigo, 2016*), whereafter local vessels increasingly targeted shortfin makos, as seen from sharply increasing CPUE indices in all areas during that period. Reported shortfin mako landings increased ten-fold between 2003 and 2005, with most of the sharks originating from expanded fishing grounds over the Agulhas Bank (South area), where shortfin makos are more abundant (*Smith, 2005*).

Apart from increased targeting, we suggest that sudden increases in CPUE indices of both species may have resulted from substantive changes made to fishing regulations and their enforcement (*Da Silva et al., 2015*; Table 2). Weak enforcement and under-reporting by local shark-directed vessels active prior to 2004 resulted in implausibly low CPUE indices at that time. Large-scale finning and discarding of carcasses in 2000 and 2001 (25–30% of blue sharks) were reported (*Petersen & Goren, 2007*) and landings were not reported to the IOTC in 2001 and 2002 (*IOTC, 2004*). Permits for shark-directed longliners were abolished in 2005, and formerly shark-directed vessels were allocated permits to catch

**Table 2  Changes in the regulations pertaining to sharks caught by pelagic longline fisheries in South African waters.**

| Year | Changes in regulatory environment | Reference |
|---|---|---|
| 1998 | • Shark finning prohibited in South Africa | *Camhi et al. (2009)* |
| 2002 | • 100% Observer coverage on foreign flagged vessels in South African waters | *West & Kerwath (2015a)* |
| 2004, 2005 | • Shark finning banned by ICCAT and IOTC | *Camhi et al. (2009)* |
| 2005 | • Swordfish and tuna longline fishery commercialised in South Africa | *Smith (2005)* |
| | • Shark-directed permits abolished, but some local vessels continue to target sharks under a permit exemption | |
| 2006 | • 10% Shark bycatch limit and release of live sharks imposed | *Clarke & Smith (2008)* |
| 2011 | • Shark vessels fishing under exemption fully amalgamated into tuna and swordfish fisheries | *Da Silva et al. (2015)* |
| | • Sharks managed as bycatch of tuna and swordfish longline fishery | *West & Smith (2012)* |
| | • Retention of whitetip (*Carcharhinus longimanus*), thresher (*Alopias* spp.), hammerhead sharks (*Sphyrna* spp.) prohibited | *Da Silva et al. (2015)* |
| 2012 | • Retention of silky sharks (*C. falciformis*) prohibited | *Da Silva et al. (2015)* |
| | • Precautionary upper catch limit of 2,000 tonnes dressed weight for sharks per year | *West & Smith (2013)* |
| | • Shark fins to be landed with trunks | *West & Smith (2013)* |
| 2014 | • Tuna and swordfish sub-sectors merged to become the Large Pelagic Longline Fishery | *West & Kerwath (2015b)* |

swordfish and tunas (*Da Silva et al., 2015*), within a stricter regulatory framework. Some vessels continued to land sharks under an exemption, however, but the sudden increase in CPUE indices in 2003–2005, and continued higher index levels thereafter in most cases, most likely demonstrates improved landings and reporting after 2004 (*Smith, 2005*).

Shark-directed vessels fishing under exemption permits were fully amalgamated into the tuna and swordfish fisheries in 2011, whereafter shark landings were managed as a bycatch of the pelagic longline fishery (*West & Smith, 2012*). A precautionary upper catch limit of 2,000 tonnes dressed weight per year was imposed in 2012 (*West & Smith, 2013*; *Anderson et al., 2015*) and shark fins had to be accompanied by trunks, with the total weight of retained fins not exceeding 8% of shortfin mako- or 13% of blue shark landed weight, respectively (*West & Smith, 2013*). Increases in the CPUE indices of both species after 2010 suggest that reporting improved after the imposition of new regulations, but the upwards trends in both shortfin mako and blue shark CPUE indices over the past decade, mainly in the West and Southwest, indicate a continued, and potentially increasing reliance of local pelagic longliners on shark landings.

## CONCLUSION

The key assumption that CPUE is proportional to abundance was not met in this study, because the modelling framework could not reconcile the influences of inconsistent

reporting and targeting on the long-term data. The CPUE indices based on logbook data were unreliable indicators of shark abundance, but when interpreted in conjunction with reported landings, by area and season, and with hindsight of market trends and regulation changes, anomalies in the indices could be plausibly explained. The scale of CPUE increments, up to an order of magnitude on a year-on-year basis, demonstrated marked effects of inconsistent reporting and targeting, driven by market value and changes in regulations. The upwards trend in shortfin mako CPUE indices over the last years of the time series suggested that they are increasingly being targeted and retained. Our study provides a reference framework for the analysis and interpretation of commercial logbook data influenced by factors external to stock abundance - often the only source of information that fisheries researchers have for assessments.

## ACKNOWLEDGEMENTS

We thank CapFish, particularly Victor Ngcongo and Willem Louw, for their advice on how the pelagic longline fishing fleet operates, and the skippers of the longline vessels for providing insight into various fishing strategies and techniques. Thanks are due to the Department of Agriculture, Forestry and Fisheries, particularly Charlene da Silva and Wendy West, for providing the landings and logbook data for the study, and for their advice. Dr. Simon Hoyle and two anonymous reviewers are thanked for advice on how to improve the manuscript.

### Funding

This work was funded by National Research Foundation (NRF) incentive fund (grant number 96309) to Johan Groeneveld, as well as through a college bursary provided by the University of KwaZulu Natal (UKZN). The funders had no role in study design, data collection and analysis, decision to publish, or preparation of the manuscript.

### Grant Disclosures

The following grant information was disclosed by the authors:
National Research Foundation (NRF): 96309.
University of KwaZulu Natal (UKZN).

### Competing Interests

The authors declare there are no competing interests.

### Author Contributions

- Gareth L. Jordaan and Johan C. Groeneveld conceived and designed the experiments, performed the experiments, analyzed the data, contributed reagents/materials/analysis tools, prepared figures and/or tables, authored or reviewed drafts of the paper, approved the final draft.

- Jorge Santos conceived and designed the experiments, performed the experiments, analyzed the data, contributed reagents/materials/analysis tools, authored or reviewed drafts of the paper, approved the final draft.

## Data Availability

The logbook data provides active fishing positions and the catches of sharks made at each, and cannot be made available in the public domain. Scientists that are interested in using the raw data for bona fide research must apply directly to the Chief Director: Fisheries Research and Development at the Department of Agriculture, Forestry and Fisheries (DAFF) in Cape Town, South Africa (email to kimp@daff.gov.za). After permission is granted by DAFF (on a case by case basis), data files can be made available.

## Supplemental Information

Supplemental information for this article can be found online at http://dx.doi.org/10.7717/peerj.5726#supplemental-information.

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
