# Peer review of "Effects of inconsistent reporting, regulation changes and market demand on abundance indices of sharks caught by pelagic longliners off southern Africa"

_PeerJ, doi:10.7717/peerj.5726_

## Round 0.1 · original submission · Major Revisions

One of the major criticisms by both reviewers (and I agree with it) is the aggregation of fishing set data. It is not clear why authors decided to do this since using the individual sets would provide a more robust analysis.
Even though the second reviewer recommended a Reject, I believe that the suggestions made by Simon Hoyle and by the second reviewer (many of them coincident) are executable and should be enough to make this paper publishable.

I realize that the changes requested require a reorganization of the data and a restart of the analysis (with a consequent rewrite of at least part of the text) but I believe the data and the relevance of the subject are worth the effort.

·

Basic reporting

The English is generally good, clear and unambiguous. There are minor grammatical errors throughout, some of which I’ve marked in the pdf, e.g. at lines 60, 65, 73, and 75.

The introduction provides appropriate background information to show the context.

The analysis code has not been provided. The raw data have been provided but are incomplete, lacking the ‘observer present’ variable. Figures are provided in the MS but not as separate files.

The data are aggregated but the data file reports a precise location for each row of data. Is this a median or average location or simply the location of the first set?

Acknowledgments include funders, which should instead appear in a separate funding statement.

Experimental design

The research appears to be suitable for the journal, and the research question addresses an important issue that is clearly identified in the MS.

The methods are not sufficiently well described to reproduce the analysis. The model equations should be supplied in the text, and the analysis code should also be provided. I loaded the data into R and ran gamms with lme4, following the reported methods as closely as possible. The delta models all failed to run.

More information should be provided about the error back-calculation and error propagation methods, rather than just giving references.

Validity of the findings

I was puzzled by the decision to aggregate the data before analysis, given that the set by set data were available. In aggregated data, the uncertainty associated with each row is affected by the number of sets aggregated, which breaks the assumption that the data are iid. This is not a major problem but could easily be avoided.

More serious is that in a binomial model, effort (hooks) should be included in the gamm as a predictor, because effort affects the probability of non-zero catches, i.e. a cell with more effort will be more likely to catch at least one shark. However, Tables S1a and S1b suggest that effort was not included in the binomial model, which will bias the standardization results.

The standardization formulae appear to include ‘- 1’ (see Table S1a), which forces the model through the origin. This is unusual and may be problematic, and should at minimum be discussed and justified.

I would like to see more data exploration, perhaps in the supplementary material, to show that data are being modelled adequately. For example, I found that for N_BSH the seasonal relationships seem to be different in each region, though they are quite small effects so probably not problematic. There is no need to fit season as a categorical variable, when month could be fitted as a continuous variable.

Please explain why all the covariates are fitted as random effects. What is the benefit? It would be useful to fit them as fixed effects, and show the parameter estimates, which could be compared with the scale of the inter-annual changes. In general, the analysis would be stronger if the CPUE standardization was done more carefully. The same covariate model seems to be applied to both species and to the binomial and continuous components, but may not be the best approach in all cases.

There are sufficient data to analyse each region separately, and doing so would strengthen the argument if each region shows the same pattern in 2010, which seems to be the case for blue shark.

The statement is made that the observer CPUE declined 2000-2005. It would be useful to plot these estimates, and compare the observer catch rates with the logbook catch rates.

The recent increase in Japanese standardized shortfin mako CPUE may also be affected by changes in reporting, so may not support the hypothesis of increased shark biomass. See chapter 1 of Hoyle et al 2017 below.
- Hoyle, S.D., Semba, Y., Kai, M., Okamoto, H. (2017) Development of Southern Hemisphere porbeagle shark stock abundance indicators using Japanese commercial and survey data. New Zealand Fisheries Assessment Report 2017/07. WCPFC Scientific Committee 13th regular session WCPFC-SC13- SA-IP-15: 64.

The discussion is generally well argued and consistent with the evidence, apart from the concerns identified above.

Reviewer 2 ·

Basic reporting

The basic reporting of the paper seems OK. But see general comments to authors and Editor below.

Experimental design

The experimental design seems to have considerable problems, as for example the use of only log-book data aggregated by month for this type of analysis. But see all details in the general comments to editor and authors below.

Validity of the findings

Given the issues identified in the validity of data, and other issues in the methods as for example not being clear why the GLMMs all variables (except year) are used as random, I have serious doubts about the validly of the results. But see the general comments to authors and Editor below.

Additional comments

Main issues:
Data: The data used seems to come exclusively from logbooks. While this is valuable data, there may be issues with reporting, species ID, recording (or not) of discards, etc. As observer data is also available in some cases (mentioned in the text), a comparison should be made between the two sources and discussed.

Representativeness: The fleets analyzed operate exclusively in the South Africa EEZ, while the species analyzed (blue shark and shortfin mako) are highly migratory (South Atlantic and whole Indian Ocean stocks). Therefore, the trends described for those local operating fleets cannot really be extrapolated to population levels. Such data could potentially provide some interesting local indicators, but even those need to be carefully discussed given the source of the data (logbooks) with all its problems and limitations.

Methods: The methods used (Delta method for combing GLMs) seem standard for CPUE standardization. However, there are issues not well explained as for example how the data was filtered and also the % 0 (I assume it was the reason why the Delta approach was chosen, but if that is not specific the reader does not know). Also, it seems from the text and the available dataset on the additional materials that the data was grouped by vessel/month which is also not clear, and can have issues of losing variability associated with set-specific data. Finally, the models used were GLMMs but the only fixed effect was "year", and the reason for formulating and using this approach is not clear. Typically, a random effect would represent variables for which only the overall variation is needed (e.g., vessel effects) while variable that are interesting to quantify directly (e.g, spatial/seasonal) should be used as fixed effects.


Further detailed comments are listed below:

The criteria used for any data filtering/cleaning should be clearly described.

At some point, and even though it seems that set-by-set data is available, this data was then grouped for the analysis. This can be a problem, as much o the variability associated with the fishing activity can be lost with such data aggregations.

Could the authors please refer if there are any possible ID issues between short and long fin makos? Isurus oxyrinchus vs Isurus paucus? I would assume that especially in logbook data this can be a serious problem.

Were interactions tested and/or used in the models? I cannot find deviance table and it would be important to have and analyze those.

The GLMMs seem that only the year was used as fixed while other variables were used as random. I would assume that variables such as spatial/seasonal effects should also be used as fixed, as it might be important to study their effects in the CPUE rather than only take into account their variability.

Were the % 0s similar, i.e., a similar issue for the 2 species? Where other models for dealing with 0s considered? As the CPUEs are considered in n/1000hooks, models for count data (e.g, Poisson, NB, Zero-inflated Poisson or Zero-inflated NB, depending on data characteristics and dispersion) could have been tested and compared.

The method used for getting the year effect is not clear, and how the coefficients were used. Typically, for CPUE standardization we want to extract the marginal means (ie.LS means) for the year effect. Was this what was done? But as it seems that the only fixed factor used was year, then in this case only the year coefficients are considered? Either way, the methodology used is not clearly described and raises questions on if it was applied correctly.

Any possibility that the discarding practices (or under-reporting rates) may have changed over time. If the discarding/reporting was consistent through time then it is only a problem in the scale. But if there are changes through time, then the entire results/trends will be biased.

The increases do not seem to be biologically possible to be explained as abundance trends. So this likely means that there are other effects (e.g, targeting, reporting, discarding vs retention, etc) that are not being taken into account in the standardization. In that case, the resulting standardized index should not be used as a proxy for abundance indicator. This is discussed alter in the discussion of the paper, but as it is not clear what is causing those trends, then the utility of the results is also not very clear.

As those increases are seen for the local fleet, it must really be either targeting and/or reporting issues that are affecting the results, which will be masking with any abundance trends in the species.

It would also be important to analyze some type of mosaic plot (or have a table with N observations per category) so that the reader can understand if there are actually data for all explanatory variable combinations. This is especially important in cases as the Delta method that has a binomial component, with assumptions in terms of samples per category.

---

## Round 0.2 · Major Revisions

I agree with Simon Hoyle's review and decision. One of the advantages of Open Science is that the methods can be repeated/reused to verify the validity of the results and/or conclusions. Using the individual set position to create 5-degree squares is common practice in many (all?) RFMOs. In the spirit of Open Science, Simon Hoyle kindly provided R code with suggestions for further data exploration which I believe can be useful to provide more robust conclusions.

·

Basic reporting

No comment

Experimental design

Regarding the use of random effects, a likelihood ratio test is not an appropriate way to choose between fixed and random effects – using random effects will always use fewer parameters and have lower BIC. There are various rules of thumb to decide which to use. E.g. see Gelman 2005. Also see https://stats.stackexchange.com/questions/4700/what-is-the-difference-between-fixed-effect-random-effect-and-mixed-effect-mode.
Monthly effects are not samples from a larger population, and they’re not random samples from a normal distribution, so would be better as fixed effects. The same applies to the region effects. Modelling vessel as a random effect is fine, but shouldn’t make much difference either way. Since you have most of the vessels I would probably use fixed effects, because then it is easier to explore the estimates which may be informative. If they are not normally distributed then normally distributed random effects is not the right approach.

There is a lot of data, so very little difference is expected between the fixed effect and random effect estimates, and the choice would hardly change the year effect outcomes at all.

The use of the large regions as the only area effects will affect the indices. The areas are very large compared to the usual practice in CPUE standardization. For example, the IOTC recommends using 5-degree squares when modelling large ocean areas. Data exploration indicates strong spatial patterns within regions, which are not adequately modelled.

In the delta model, the trend will be affected by the chosen mean probability level. How did you choose the baseline levels of the categorical variables? These often need to be adjusted so that the mean of the predicted values is close to the mean of the observed values.

It would be useful to model your data independently by ocean or region, since you discuss the trends of the Atlantic and Indian Ocean stocks by ocean.

Also, if you model by regioins, and find the same pattern in separate regions in which population trends are probably different, it reinforces the pattern it is due to reporting rather than population change.

Validity of the findings

I checked the results by running some analyses myself, for blue shark in the local fishery. I found a number of problems.
- There were large differences in the month effects between the regions. Best to model the regions separately.
- More importantly, there were also differences in the annual trends. This may undermine the conclusions of the paper.
- The regions were not separated appropriately in the dataset. Plotting the ‘regions’ by lat and long shows many sets in the wrong region.
- As above, there are strong spatial patterns within regions.

I recommend resubmission after rerunning the analysis to resolve these problems. I attach my data exploration code for interest.

The paragraph starting at line 378 compares ICCAT abundance trends with those observed here. The ICCAT data include South African data, which should be explicitly excluded from the comparisons.

Line 394, the issue is not if they are retained, but if they have been consistently retained.

Line 401, vessels can change their fishing strategy, so vessel is not a catch-all for targeting strategy. Has this been considered? The original dataset apparently contained information on set times and bait type, as well as number of hooks, which could be used to distinguish targeting strategy.

Line 435 states that local vessels landed far more sharks during the warmer season. But this is based on raw numbers, because there were no estimates of the covariates due to the use of fixed effects.

Line 451, sharp increases in blue shark catches were far less apparent when the data were analysed separately by region, with a model that accounted better for spatial variability. The same may be true for SMA.

# Exploration code
library(mgcv)
library(maps)
library(maps)

dat <- read.csv("peerj-21432-Raw_Data_Individual_Set_Data_GLMM.csv")

dat$lat1 <- floor(dat$LAT)
dat$lon1 <- floor(dat$LONG)
dat$vess <- factor(dat$VESSEL)
dat$yr <- factor(dat$YEAR)
dat$flt <- factor(dat$FLEET)
dat$ssn <- factor(dat$SEASON)
dat$ar <- factor(dat$REGION)

str(dat)

dat$N_BSH = dat$BSH * dat$EFFORT / 1000
dat$N_SMA = dat$SMA * dat$EFFORT / 1000
dat$HOOKS = dat$EFFORT

loc <- dat[dat$FLEET == "LOCAL",]

mod <- gam(log(N_BSH + 1) ~ yr + s(MONTH) + te(LONG, LAT, k = c(20, 20)) + HOOKS + vess, data = loc, family = gaussian)
windows()
par(mfrow = c(2,2))
gam.check(mod) # this simplified model fits well enough for basic inference, though a delta model would be better
AIC(mod)
plot.gam(mod, pages = 1, all.terms = TRUE, scheme = 2)
table(dat$ar)
summary(mod)

windows(10,8)
plot.gam(mod, select = 2, ylim = c(-40, -25), scheme = 2, too.far = .01)
map(add=T, fill = T)
points(loc$LONG, loc$LAT, cex=0.2)

mod <- gam(log(N_BSH + 1) ~ yr + s(MONTH) + te(LONG, LAT, k = c(20, 20)) + HOOKS + vess, data = loc, family = gaussian)
windows()
par(mfrow = c(2,2))
gam.check(mod) # shows that this simplified model fits well enough
AIC(mod)
plot.gam(mod, pages = 1, all.terms = TRUE, scheme = 2)
table(dat$ar)
summary(mod)

windows(10,8)
plot.gam(mod, select = 2, ylim = c(-40, -25), scheme = 2, too.far = .01)
map(add=T, fill = T)
points(loc$LONG, loc$LAT, cex=0.2)

mod <- gam(log(N_BSH + 1) ~ yr + ssn + te(LONG, LAT, k = c(10, 10)) + HOOKS + vess, data = loc, family = gaussian)
summary(mod)

mod_w <- gam(log(N_BSH + 1) ~ yr + s(MONTH) + te(LONG, LAT, k = c(10, 10)) + HOOKS + vess, data = loc[loc$ar %in% c("W"),], family = gaussian)
mod_sw <- gam(log(N_BSH + 1) ~ yr + s(MONTH) + te(LONG, LAT, k = c(10, 10)) + HOOKS + vess, data = loc[loc$ar %in% c("SW"),], family = gaussian)
mod_s <- gam(log(N_BSH + 1) ~ yr + s(MONTH) + te(LONG, LAT, k = c(10, 10)) + HOOKS + vess, data = loc[loc$ar %in% c("S"),], family = gaussian)
mod_e <- gam(log(N_BSH + 1) ~ yr + s(MONTH) + te(LONG, LAT, k = c(10, 10)) + HOOKS + vess, data = loc[loc$ar %in% c("E"),], family = gaussian)

summary(mod_w)
summary(mod_s)
summary(mod_sw)
summary(mod_e)

windows(12,10); plot.gam(mod_w, pages = 1, all.terms = TRUE, scheme = 2, main = "W", too.far = 0.01)
windows(12,10); plot.gam(mod_sw, pages = 1, all.terms = TRUE, scheme = 2, main = "SW", too.far = 0.01)
windows(12,10); plot.gam(mod_s, pages = 1, all.terms = TRUE, scheme = 2, main = "S", too.far = 0.01)
windows(12,10); plot.gam(mod_e, pages = 1, all.terms = TRUE, scheme = 2, main = "E", too.far = 0.01)

---

## Round 0.3 · accepted · Accept

Congratulations on the improved analysis and rewritten manuscript. This is a much improved article. I believe it would be nice to acknowledge Simon Hoyle's contribution (though this is not a requirement). I also noticed that the reference to Coelho et al, 2018 is missing some authors so I advise you to review all references carefully (you can do this while in production)

I apologize for any delays in publishing due to my availability

#